# Socially assistive robotics and older family caregivers of young adults with Intellectual and Developmental Disabilities (IDD): A pilot study exploring respite, acceptance, and usefulness

Ling Xu[1]*, Noelle L. Fields[1], Julienne A. Greer[2], Priscila M. Tamplain[3], John C. Bricout[4], Bonita Sharma[5], Kristen L. Doelling[6]

1 School of Social Work, University of Texas at Arlington, Arlington, TX, United States of America, 2 College of Liberal Arts, University of Texas at Arlington, Arlington, TX, United States of America, 3 Department of Kinesiology, University of Texas at Arlington, Arlington, TX, United States of America, 4 School of Social Work, University of Minnesota, Twin Cities, St. Paul, MN, United States of America, 5 College for Health, Community & Policy, University of Texas at San Antonio, San Antonio, TX, United States of America, 6 University of Texas at Arlington Research Institute (UTARI), Fort Worth, TX, United States of America

* lingxu@uta.edu

**Data Availability Statement:** The data underlying the results presented in this study are available

## Abstract

### Introduction

The need for caregiver respite is well-documented for the care of persons with IDD. Social Assistive Robotics (SAR) offer promise in addressing the need for caregiver respite through 'complementary caregiving' activities that promote engagement and learning opportunities for a care recipient (CR) with IDD. This study explored the acceptability and usefulness of a SAR caregiver respite program responsive to feedback from both young adults with IDD and their older family caregivers (age 55+).

### Method

Young adults with IDD and caregiver dyads (N = 11) were recruited. A mixed methods research design was deployed in three phases: Phase I with four focus groups to inform the program design; Phase II for program demonstration and evaluation with pre- and post-surveys; and Phase III with post-program interviews for feedback and suggestions.

### Results

Both young adults with IDD and their caregivers scored favorably the social presence of, social engagement, and satisfaction with robot Pepper. Though there was no significant improvement of caregiving burden/stress as well as well-being of the young adults with IDD based on surveys, results from interviews suggested that the SAR may offer physical/emotional respite to caregivers by providing companionship/friendship as well as promoting independence, safety/monitoring, and interactive engagement with children.

from the Texas Data Repository (https://dataverse.tdl.org/dataverse/uta). Specifically, data for this paper are shared at: DOI: 10.18738/T8/ZU0XVW, URL: https://doi.org/10.18738/T8/ZU0XVW.

**Funding:** The contents of this article were developed in part under a grant from the National Institute on Disability, Independent Living, and Rehabilitation Research (NIDILRR), Wireless Inclusive Technologies Rehabilitation Engineering Research Center (grant number 90RE5025), and the Advanced Rehabilitation Research Training (ARRT): Inclusive Technology and Policy Design Research Fellowships (grant number 90ARPO0002). NIDILRR is a Center within the Administration for Community Living (ACL), Department of Health and Human Services (HHS). The contents of this paper do not necessarily represent the policy of NIDILRR, ACL, HHS, and you should not assume endorsement by the Federal Government. The funders had no role in study design, data collection and analysis, decision to publish, or preparation of the manuscript.

**Competing interests:** The authors have declared that no competing interests exist.

## Discussion

SAR has potential in providing respite for older family caregivers. Future studies need a longer program design and larger sample size to develop a promising intervention and test its feasibility and efficacy.

## Introduction

The percentage of people with a disability was 12.7% in 2017 [1] and up to 3% of the US population (7 to 8 million Americans) have a disability identified as an intellectual and developmental disabilities (IDD) [2]. Approximately 39.8 million caregivers (16.6% of Americans) provide care to young adults (aged 18+) with a disability or illness [3]. Specifically, there are roughly 9.3 million caregivers who are providing care to a child or an adult with some kind of disabilities under the age of 50 [4]. Families remain the primary caregivers for adults with developmental disabilities with roughly 76% of individuals with IDD residing at home [4]. In 25% of these homes, the family caregiver is over 60 years of age, and the average age of their care recipients (CRs) with a developmental disability is age 38 [4].

Caring for an adult child with IDD presents multiple stressors and challenges, accompanied by high levels of anxiety, depression and stress [5]. In addition to caregiving-related stress, older family caregivers of IDD also face their own ageing process and health problems, which may further worsen their overall health conditions [6]. These stressors highlight the need for and value of respite care (short-term relief or break) for older family caregivers. The use of technology for respite care is an emerging area of research. The urgency to advance artificial intelligence (AI), namely, humanoid robotics, to help with caregiving [7] is timely and significant for this population.

A recent systematic review on socially assistive robots (SAR) providing health and social care for older adults [7] found only few studies, and most importantly, none investigated respite care. With an ageing population, respite care is recognized as needed, but insufficiently available. Support for older adults and their families offers psychological, social and quality of life benefits to both caregiver and CR [8–10].

Additionally, young adults with IDD are very likely to experience prejudice and discrimination in the community [11] and experience social exclusion [12]. The absence of social inclusion may lead to risky behavior, increase the risk of physical, psychological and sexual abuse, and have a negative effect on health and well-being of young adults with IDD and their family caregivers [13, 14]. Spending more time with peers and in institutional contexts, staff, rather than parents are an important component of social inclusion and development for young adults with IDD [15].

In this study, we piloted a design and implementation of a SAR respite program for older family caregivers (age 55+) who cared for young adults with IDD and evaluated its potential for reducing caregiving burden/stress through structured interactions with the robot Pepper, together with a video narrative, aimed at increasing levels of engagement, companionship, and learning of young adults with IDD. While not a peer or a staff member, Pepper, a wheeled humanoid robot designed to operate in a human-centered environment [16], was viewed as a social agent (and therefore a third-party being) by both the young adults and their families, expanding their social circle. Therefore, Pepper had a role to play in terms of perceived companionship and support for the young adults with IDD, with the caveat that socially assistive robots are complementary to, and not in substitution of, human companionship [17].

## Respite programs or intervention for family caregivers of persons with IDD

Family caregivers of persons with IDD tend to be vulnerable to physical strain and health risk behaviors, psychological problems (e.g., depression and anxiety symptoms), negative changes of family relationship and social network, and financial hardship due to out-of-pocket expenses and employment-related costs [18–20]. Research also suggests that stress on a parent caregiver can have an effect on the caregiver's physical health, along with a lack of "mastery" in their ability to provide care for their child [21]. Most programs designed to address the mental health of caregivers come in the form of stress reducing intervention. For example, a community-based intervention called Mindfulness Based Stress Reduction (MBSR) used mindfulness and demonstrated decreased stress and anxiety among caregivers [22, 23].

Respite care is an important program or intervention for family caregivers, because they can decrease stress in caregivers of persons with IDD and reduce mental health symptoms (i.e., depression or anxiety) [24, 25]. Research has also shown that many families choose to find respite services outside of the home or by combination of other services [25]. In-home respite services are starting to become more available, and there may be some financial supports available for these services [26]. These financial supports can come from Medicaid or other forms of ongoing health insurance provide to children and youth with special needs. However, it is evident that gaps remain regarding what we know and how we can support respite care for caregivers of young adults with IDD.

## SAR and respite care for family caregivers of persons with IDD

As an advanced form of AI, humanoid robotics have shown promise in the literature related to children with disabilities as well as older adults. SAR, also known as social robots, interact with humans across a variety of settings, and engage people in learning, social, rehabilitative, assistive care, and collaborative activities [27–30]. Furthermore, SAR promotes independence and well-being while interacting with the user in an intuitive fashion without extensive training or the intervention of a human operator [31]. An interaction between a child and a robot has positive effects on children with IDD and may have a significant influence in a child's social interaction, while there are also some increases in a child's vocabulary and motor imitation [32]. Assistive technologies like SAR are being considered as enablers to support the process of caregiving [33]. For example, therapeutic SAR improved mood and stimulated social interaction and communication in the care of older people with dementia in Hong Kong [34], as well as provided sensory enrichment and social connectivity to persons with dementia and respite to partners in Australia [35]. However, the use of SAR as a tool for respite, especially for older family caregivers of young adults with IDD remains under-studied. Among the limited literature, potentially positive outcomes were achieved through SAR in Australia in terms of engagement, productivity and usefulness as well as reciprocity of people with autism spectrum disorder and respite to their caregivers (e.g., parents) in their day-to-day living [35].

Given these gaps in the literature, rapidly changing demographics in the United States, and the emerging socio-technological landscape worldwide, it is a critical time for researchers to seek programs or interventions to promote the wellness of older caregivers and to engage their young adult children with IDD through arts that inform the design of cutting-edge technology in the guise of social robots. Therefore, this study developed and delivered an intervention and tested its effectiveness from the young adult-caregiver dyads' perspectives.

## The present study

To address the abovementioned literature gap, the present study aimed to develop a caregiving model that would provide caregiver respite to relieve caregiving stress during times when the

caregiver could not be physically co-located with the care recipient (i.e., young adult with IDD). The caregivers and care recipients in this study are co-community dwelling, in which the caregiver is the parent or grandparent, and the care recipient is the young adult with IDD. Thus, we will refer to the care recipient as a young adult with IDD, acknowledging that they live together in a home/familial setting.

This pilot (preliminary) study had the goal of developing a method for the SAR to assume respite roles for caregivers of young adults with IDD, by using "Pepper". The primary goal of this study is to test if a SAR (Pepper) can help relieve caregiving stress/burden by providing respite to older family caregivers through the provision of a complementary and temporary caregiver role. The secondary goal is to test the acceptance and usefulness of a SAR approach in providing respite. Specifically, the aims were: (1) the development of SAR that was well accepted by both young adults with IDD and their older family caregivers; (2) the development of adequate responsiveness and engagement with young adults with IDD; (3) potential for improving the wellbeing of young adults with IDD, and (4) potential to provide respite and improve the wellbeing of older family caregivers.

## Methods

### Study design

We used a mixed methods experimental quantitative design [36] (see Fig 1), in which we collected the qualitative data before the program (Phase I, focus groups) to help with the development of the SAR program (Phase II) as well as after the program (Phase III, individual interviews) to enrich the interpretation of the experimental results. We collected quantitative data using survey in Phase II program demonstration. Qualitative data can lay a strong foundation for subsequent pilot program work by facilitating the development of an underlying study conceptualization [37]. Qualitative methods can also be particularly useful when data or information is limited, and a greater understanding is desired after the pilot program and/or for collecting information on the acceptance and usefulness of a new technology [38].

### Study participants

This specific study was reviewed and approved by the Institutional Review Board (ethics committee) of the University of Texas at Arlington before the study began (approval number: 2018–0203). Participants were recruited with the support of the CEO and staff of a community partner (i.e., Helping Restore Ability, HRA) that serves persons with disabilities and their

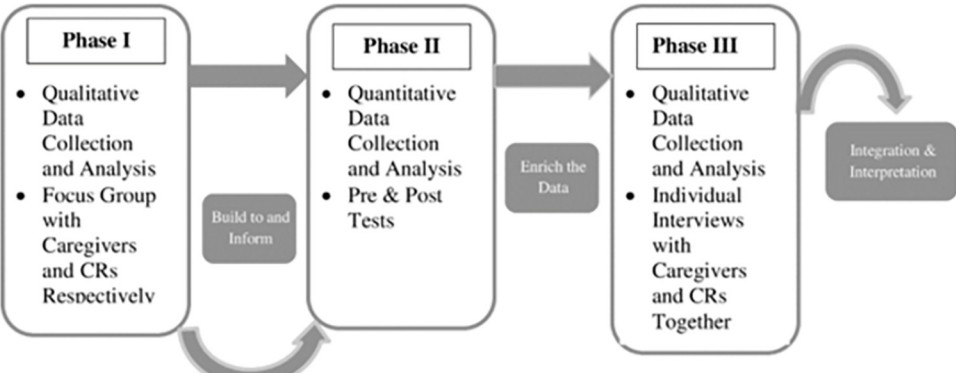

**Fig 1. Mixed methods experimental quantitative design of the study (adapted from Creswell and Plano Clark, 2017).**

families. With information provided by the agency, the research team then contacted potential participants to explain the study in greater detail. If individuals stated interest in participating, an in-person meeting was then scheduled with the research team to sign the consent form (written) and conduct pretest survey. During the physical meetings, the researcher went through the consent forms with young adults with IDD and their older caregivers, line by line, stopping to make sure they understand. Some (approximately half) of the young adults with IDD needed to understand the consent form with the help of their older family caregivers. After both young adults with IDD and their older caregivers signed the consent forms in written format respectively, pre-tests were completed for them respectively while the researcher read the questions to them and wrote down the answers.

The inclusion criteria for the family caregiver were: (1) 55+ years old; (2) live-in family member or relative to CR; (3) absence of cognitive impairment that would prevent them from consenting to the study (tested by the Six-Item Screener (SIS) [39]); and (4) speak English. The eligibility criteria for the young adults with IDD included that they: (1) had some form of IDD; (2) 18+ years of age; (3) absence of cognitive impairment that would prevent them from consenting to the study (same screening tool as caregivers); and (4) speak English. A total of 11 dyads consented to participate, completed the pretest, and participated in the focus group and program demonstration. The descriptive of the dyad samples are shown in Table 1.

### Data collection

**Phase I: Focus groups.** The purpose of Phase I was to gather feedback and information from the family caregivers and young adults with IDD about the use of the social robot, Pepper, for a respite program. The overall intent of Phase I was to gather input from the dyads in order to design and develop the program. Four focus groups (two for caregivers and two for young adults with IDD) (*n* = 11 dyads) were arranged on two different days, during which a custom video was shown about the SAR capabilities [40]. A hard copy picture of Pepper was also offered to the dyads to take home. Subsequent to watching the video, questions about SAR-enabled care systems were discussed among caregiver and young adult groups (separately). Focus groups for caregivers lasted about 1 hour while focus groups for young adults with IDD lasted about 30–45 minutes. The focus groups took place in two private rooms at HRA. In general, probe questions for older family caregivers were centered on how they thought about respite care, how they felt leaving young adults with IDD with a SAR, what benefits and challenges they thought of a SAR helping them with respite care. For young adults with IDD, questions were mainly about what contributed to their overall well-being and how, what kinds of activities that a SAR (shown before the question) might offer that they would find enjoyable, and/or positively impact their mood.

**Phase II: Program demonstration and pre/post survey.** After listening to the participants' feedback and expectations about social activities of the SAR in respite care, the research team programmed Pepper. The interactions programmed were mainly social activities, such as reading a story, playing fun activities, dancing together, etc. A beta test of the robot program was arranged with an 8-year-old volunteer with no disabilities (permission granted by parent). After revising the programing based on observations of the beta test as well as feedback from the volunteer, the full demonstration of the program was applied to the participants. The 11 young adults with IDD interacted with Pepper individually and each demonstration took approximately 15 minutes, while their older family caregivers took respite in another room where a video was set up for them to observe their child's interaction. Pre and post surveys were administered by trained interviewers before and after the program for caregivers and their young adults with IDD respectively. Each survey took approximately 10–30 minutes.

**Table 1. Descriptive of the dyads samples (*n* = 11 dyads).**

| Variables | % | Mean (SD) | Range |
|---|---|---|---|
| **Older Caregivers** | | | |
| Age | | 60.21 (4.49) | 55–68 |
| Gender | | | |
| Male | 14.3 | | |
| Female | 85.7 | | |
| Marital Status | | | |
| Married and cohabitating | 71.4 | | |
| Divorced | 28.6 | | |
| Living Arrangement | | | |
| With this child and spouse | 71.4 | | |
| Only with this child | 28.6 | | |
| Education | | 7.43 (1.09) | 1–9 |
| ADLs | | 6.50 (1.34) | 6–11 |
| IADLs | | 4.71 (1.07) | 4–7 |
| **CRs with IDD** | | | |
| Gender | | | |
| Male | 57.1 | | |
| Female | 42.9 | | |
| Age | | 26.57 (7.99) | 18–42 |
| Living Arrangement | | | |
| Live only with parents | 92.3 | | |
| Live with others | 7.7 | | |
| Education Level | | | |
| Middle school (grades 6–8) | 7.1 | | |
| High school (grades 9–12) | 14.3 | | |
| High school graduate or equivalent | 78.6 | | |
| Self-rated Health | | 3.86 (1.29) | 1–5 |
| Satisfaction with Health | | | |
| Very unsatisfied | 7.1 | | |
| Fair | 21.4 | | |
| Satisfied | 28.6 | | |
| Very satisfied | 42.9 | | |
| ADLs | | 9.0 (4.26) | 6–18 |
| IADLs | | 6.0 (3.11) | 4–12 |

SD = standard deviation

The program demonstration took place inside the Emotional Robotics Living Lab in the author's institution, a 'research room' space that is dedicated to providing a living area for research studies to simulate the home-space of participants. In this way, participants receive a home-like environment while interacting with the social robot and reimagine how that robot might interact with their real home life. Upon arrival, the young adult with IDD and the caregiver were taken to a waiting room with a team member to greet them which also gave participants relaxation and break time before the demonstration. The young adult was then brought to the lab by a team member and the caregiver stayed in another room to observe the study. The study room and the observation room were adjacent in the building. Each team member in the lab was introduced to the young adult before the start of the study. Three members of

the interdisciplinary team remained in the room who stayed in the corner of the room and did not interfere with the demonstration process. One member introduced the robot to the young adult. A second team member provided the programing instructions for the robot, and a third team member who had expertise in working with children with IDD provided potential support to the young adult participant, if needed, during the demonstration.

After introducing Pepper to the young adult, the team allowed time for the young adult to become familiar with the robot by touching Pepper and asking a few questions. The team then outlined a space where it was safe for the young adult to stand near the robot. A programmed physical greeting of a fist bump began the interaction. The young adult was then guided by Pepper to move toward and sit on the sofa and listen to a story that highlighted resiliency, friendship, and human-machine potential for companionship. While Pepper recited and performed the story to the young adult, music was added to tonally create a sound atmosphere that mirrored the emotional concepts in the story. Interactive engagement was programmed at the end of the original narrative, when the young adult was asked by Pepper to share their own story about school or any open-ended responses to Pepper's story. Pepper asked the young adult to move again to the center of the room to give the young adult room to move after the story engagement. The following exercises and dances were programmed to interact with the young adult and to carry-out a pre-study focus group caregiver request for movement and exercise for the young adult. A theatre mirroring exercise was performed to engage the young adult with the robot and companionship. Music, in the form of air guitar and saxophone riffs played next to encourage the young adult to join in the dancing with the robot. A rhythmically slower tai-chi exercise finalized the movement portion of the programming engaging the young adult while taking care not to overstimulate or scare the young adult. The robot was finally programmed to reflect enjoyment of the time spent together with the young adult, wave to the young adult, and say goodbye. At the end of the demonstration, the caregiver was invited into the research room and take pictures with the young adult and the robot. Both caregiver and young adult were lastly guided to a final room to complete post tests and interview to document their experiences interacting and observing the program.

**Phase III: Follow-up qualitative interviews.** After the program, the caregiver and young adult as a family dyad were interviewed in another room, which lasted approximately 30–45 minutes. Interview guiding questions were used (see Table 2). The interview asked dyads'

Table 2. Phase III interview guild.

| For CRs with IDD: |
| --- |
| 1. How do you feel about your interaction with the robot? How did you feel about being with the robot while your caregiver is not present? |
| 2. How did you feel about the amount of time you and the robot spent together? |
| 3. How do you think this robot might help you with your mood? |
| 4. If you were home, instead of here, how do you think you would feel differently about the robot being with you? |
| 5. What suggestions do you have for this project or for interacting with a robot for a brief time without your caretaker? |
| **For Older Caregivers:** |
| 1. How comfortable were you leaving the room after the robot was introduced? Why? |
| 2. If you were home, how do you think your feelings would be different, than if you were here? Why? |
| 3. Do you think the robot Pepper could offer greater independence to your CR? |
| 4. During your away from your CR, if the robot helped provide you with time and space to briefly "recharge," do you think you are better off with a respite robot? Why? |
| 5. What suggestions do you have for this project? or having a robot provide you with a break or respite? |

opinions about the social presence of Pepper, the potential to provide respite care, as well as suggestions for future programming/program.

## Measurements

The perception of robot (measured after the program demonstration) as well as the well-being for caregivers and their young adult CRs (measured before and after the demonstration) were quantitatively measured only in Phase II. The measurements of perception of robot included social presence of robot, social engagement with robot, perceived responsiveness of robot, satisfaction with robot, and perceptions of robot (only for caregivers). *Social presence of robot* was measured by 7 items that asked both caregivers and their young adult CRs to use a number 1 to 10 to indicate their feelings about the interaction with Pepper [41]. These feelings included interacting with an intelligent being, accompanied with an intelligent being, alone, attention to, involvement, responding to, and communicating to each other. *Social engagement with robot* was measured by one question asking both caregivers and their young adult CRs the level of engagement between young adults and Pepper during the program with 0 = *no engagement* and 5 = *intense engagement*. *Perceived responsiveness of robot* was measured by General Responsiveness Scale that has been validated and found reliable in prior studies [42, 43]. The measurement assessed perceptions of how understood, validated, and cared for the participants felt when interacting with the robot. Both caregivers and their young adult CRs rated nine statements using a 5-point Likert scale (1 = *not at all*, 5 = *completely*). *Satisfaction with robot* was measured by asking both caregivers and their young adult CRs four questions about levels of satisfaction in meeting their needs, relationship with the robot, meeting original expectation, as well as problems interacting with the robot (1 = *very unsatisfied*, 5 = *very satisfied*). Moreover, *perceptions of robot* was measured (only for caregivers) using Godspeed Questionnaire Series (GQS). GQS is one of the most frequently used questionnaires in the field of Human-Robot Interaction [44]. GQS consists of five scales that are relevant to evaluate the perception of (social) Human-Robot Interaction. The scales cover anthropomorphism (5 items), animacy (6 items), likeability (5 items), perceived Intelligence (5 items), and perceived safety (3 items for safety at the beginning and towards the end respectively). Each item was measured by five-point semantic differentials representing "least" to "the most".

The well-being outcome of older caregivers included self-related health and caregiving stress/ burden. Self-rated health was measured by a single item asking in general how the current health was (from 1 = *poor* to 5 = *excellent*). The Short Form Zarit Burden Interview (ZBI-12) was used to test caregiver stress/burden [45]. The 12 items address the perceived impact of the act of providing care on the physical health, emotional health, social activities and financial situation of the caregiver. Each item has five response options ranging from "0 = *never*" to "4 = *nearly always*". The ZBI-12 has shown good validity and reliability coefficients among older family caregivers [46]. Sum scores were created with higher scores indicating higher levels of caregiver stress/burden.

For the well-being of young adults with IDD, the Quality of Life (QoL) was measured through QoL Integral Scale [47]. A total of 24 items were asked that included emotional (4 items), material (7 items), physical (7 items), and social (6 items) well-being. Each item was measured with 4 responses ranging from "*1 = strongly disagree*" to "*4 = strongly agree*". The psychometric properties of the QOL Integral Scale indicate that the instrument is reliable and valid among persons with IDD [48]. Sum scores for each well-being were calculated with higher score indicating higher level of well-being.

## Data analyses

*For quantitative data analyses in Phase II*: We first conducted a univariate analysis to describe the demographic characteristics of the caregivers and the young adult CRs, as well as the key

outcome variables. Analysis of Wilcoxon Signed Ranks was conducted to assess whether the main outcome variables differed after the program demonstration. This analysis took into account the paired samples and small sample size, and therefore was appropriate for the present study.

*For qualitative analyses in phases III*, the interviews were transcribed verbatim. Conventional content analysis [49] was used to analyze the data by two members of the research team. Following the guidelines recommended by Hsieh and Shannon [49], the researchers who had not conducted the interviews first read through the transcriptions in order to gain a general understanding of the interview content. Next, the researchers read through 25% of the transcripts again and developed codes and working definitions for each code [50]. As part of this process, the researchers discussed any disagreements until consensus was reached. The researchers independently coded the remaining 75% of the transcripts and then met to group the codes into categories of related meaning. As part of the analysis, the researchers avoided using preconceived categories and allowed the categories to come directly from the data [50]. As part of establishing trustworthiness, the data were triangulated with a third researcher in order to better understand the emergent themes and to avoid bias. Peer debriefing was also used to enhance credibility [51] as well as through referencing field notes and observations from the focus groups.

## Results

Phase I focus group interview was used to inform the SAR programming. Caregivers and their young adult CRs offered insight and recommendations regarding respite care and the activities of the social robot, the majority of which were taken into consideration for the program design. Here we only reported the feedback from the dyads from Phases II and III.

### Phase II: Quantitative surveys before and after the program demonstration

**Perceptions of robot.**   Only 8 dyads had time to conduct the posttest and interview immediately after the program demonstration because of time concern or other arrangement. Both older caregivers and their young adult CRs had favorable scores of the social presence of robot Pepper (*Mean* = 44.25 for caregiver, *Mean* = 59.17 for young adults, *Range* = 7–70). For social engagement, the older family caregivers reported the maximum score of 5 while young adults thought their participation was 4.8 out of 5 (see Table 3). Dyads also reported favorable scores of perceived responsiveness of robot Pepper (mean scores were 20.8 and 26.0 out of 30 for caregivers and their young adult CRs respectively) as well as their satisfaction with robot Pepper (*Mean* = 14.25 for caregivers, and *Mean* = 16.17 for young adults, *Range* = 5–20). In addition, older caregivers reported specific perceptions of Pepper in terms of anthropomorphism (*Mean* = 19.4, *Range* = 5–25), animacy (*Mean* = 24.60, *Range* = 5–30), likeability (*Mean* = 25.00, *Range* = 5–25), perceived intelligence (*Mean* = 20.80, *Range* = 5–25), and perceived safety (*Mean* = 14.80, *Range* = 3–15).

**Dyads well-beings.**   Results from Paired Wilcoxon Signed Ranks (not shown in this manuscript) showed that caregiving burden/stress and self-reported physical health did not improve significantly after the program demonstration. At posttest, older family caregivers reported relatively medium levels of burden/stress (*Mean* = 28.17, *SD* = 4.96, *Range* = 0–48). Similarly, for the young adults, the emotional, material, physical, and social well-being scores improved somewhat, but not significantly. In general at posttest, they had relatively high levels of emotional well-being (*Mean* = 13.57, *SD* = 1.78, *Range* = 4–16), material well-being (*Mean* = 22.85, *SD* = 4.19, *Range* = 7–28), physical well-being (*Mean* = 22.57, *SD* = 2.95, *Range* = 7–28), and social inclusion (*Mean* = 17.91, *SD* = 3.44, *Range* = 4–24).

**Table 3. Descriptive at posttest (*n* = 8 dyads).**

| Variables | Mean (SD) | Range |
|---|---|---|
| **Older Caregivers** | | |
| Social Presence of Robot | 44.25 (6.29) | 7–70 |
| Perceptions of Robot | | |
| Anthropomorphism | 19.40 (4.16) | 5–25 |
| Animacy | 24.60 (1.95) | 5–30 |
| Likeability | 25.00 (0.00) | 5–25 |
| Perceived intelligence | 20.80 (2.17) | 5–25 |
| Perceived safety-at the beginning | 14.60 (0.55) | 3–15 |
| Perceived safety-at the end | 14.80 (0.45) | 3–15 |
| Social Engagement | 5.00 (0.00) | 0–5 |
| Perceived Responsiveness | 20.80 (4.09) | 5–30 |
| Satisfaction with Robot | 14.25 (0.96) | 5–20 |
| **CRs with IDD** | | |
| Social Presence of Robot | 59.17 (6.65) | 7–70 |
| Social Engagement | 4.80 (0.45) | 0–5 |
| Perceptions of Robot | 26.00 (3.58) | 6–30 |
| Satisfaction with Robot | 16.17 (3.71) | 5–20 |

## Phase III: Qualitative individual interview after the program demonstration

Participants were interviewed after the program. Several participants reported that Pepper could be an addition to their family, or that they just loved 'hanging out' with the robot. Content analysis from the interviews suggested that the SAR may offer physical and/or emotional respite for caregivers by providing companionship/friendship as well as promoting independence, safety or monitoring, and interactive engagement with their children.

**Theme one: Physical and/or emotional respite.** Even though this 15-minute short program could not clearly offer the full benefits of respite, the majority of caregivers expressed that this type of program could potentially offer them both physical and emotional respite to relieve their stress if they had a Pepper in their home. Some caregivers thought that they could take a break or do some of their favorite activities when Pepper could do the things they did for her young adult CR in daily routine:

> "That's great. You [young adult CR] would listen to Pepper better than you would listen to me! Yeah, or even mum could take a hike...I think I could rely on Pepper; if Pepper was programmed to help D [young adult CR] to do laundry, make food, then yeah. If D would listen to the robot, it would be helpful because it's the robot reminding D that he needs to do his laundry rather than me, nagging at D. So, when I say it, it's nagging. Whereas, if the robot is reminding him, then he might listen better."

> "Yes, because she's pretty dependent on me. And maybe she'd be more independent because you know, Pepper is not human. It could be like an extension of her, sort of like her...I could read a book or watch something I want to watch on TV" (Family 11 caregiver, female).

Several caregivers shared that Pepper had the unique benefit of technology that can overcome the limitations of human caregivers. As one parent mentioned "I cannot be as a caregiver, doing everything" (Family 4 caregiver, male). Other parents also said that "...it's harder

and harder as time goes by for us to interact at a comfortable level." (Family 5 caregiver, female). "I don't have a lot of time to do one-on-one or to hang. . .I thought that's just kind of a perfect thing with Pepper. . .gives him some one-on-one with somebody other than me" (Family 6 caregiver, Female). Parents reported that Pepper could provide different interactions with their young adult CRs, which provided them with a deep feeling of emotional respite. Other caregivers had similar comments:

> "So, so, so heartwarming. I think so because it would give us a, uh, like a mental. . . break. You know it really would feel comforting, you know, a real emotional help for us because when we are not around anymore, so to have something like Pepper or something like that in the future" (Family 1 caregiver, female).

> "I could do what I needed to do without feeling guilty. I could see definitely how it would be beneficial and like a lot of times I just need, you know to run up to Kroger. . .I think it would be very, very good because a lot of times, if I have things to be doing and, and he is wanting to talk to me, and ask me questions and stuff and I'm like, 'just a minute, just a minute'. So, I would feel more comfortable you know being able to get some of those tasks done if the robot was in the home and he was kind of taken care of. . .Parents get tired and cranky. . .so that would you know, be a big benefit and to help with that. . .and, and take that pressure or stress or whatever" (Family 2 caregiver, female).

> "And it allows peace of mind. I can walk out of the room, I can walk out of the house, I can even run to the store without worrying about 24/7. And now, yes, is she okay if something happens? Does she know how to call? But Pepper is there and could fill. . .It takes so much weight. That's the value" (Family 4 caregiver, male).

**Theme two: Companionship or friendship.** Companionship that caregivers cannot always provide was desired. Both caregivers and their young adult CRs mentioned that Pepper could provide companionship or friendship to the CRs. Based on his/her own experience, one young adult said, "I just really loved hanging out with the robot today!. . .And so it would be like having a friend, a buddy to talk to" (Family 2 young adult, male). Based on the observation, one caregiver shared, "I was in tears seeing the potential of this, being the companion that me, that I, her father, cannot" (Family 3 caregiver, male). Similarly, another caregiver echoed this by saying, "it would definitely help her more with having a companion there. . .Be like having somebody with her" (Family 7 caregiver, female). Other caregivers expressed similar opinions that Pepper could be friend of the young adults with IDD and provide companionship and friendship to them if they had a Pepper at home. For example:

> "And I think, uh, uh, Pepper would be great, you know for her to, to talk to, you know, a friend she uh, she is home schooled so there is not that much. . .we have social interactions but uh you know uh. . .it would be great for her to uh, to, you know share stories you know, she liked that" (Family 1, Female).

**Theme three: Promote independence.** With assistance of Pepper, caregivers found potential in promoting the independence of the young adults if they had a Pepper at home in the future. One caregiver mentioned, "I think it would help her with, I believe her self-confidence and. . .independence" (Family 1 caregiver, female). One caregiver thought programming Pepper would be prompting him [young adult CR] to make healthy choices (Family 3 caregiver, female). Another caregiver thought Pepper has potential to encourage communications with the young adult with IDD and thus promote young adult's independence.

"It would give him an opportunity to be able to communicate. . .It would help build some of his communication skills which would help with a little more independence" (Family 6 caregiver, male).

Similarly, some caregivers thought that Pepper could remind the young adults with IDD to do something that they might forget and thus helping them gain more independence. For example, Pepper could remind the young adults to take some medicine, "'Hey E, it's time to take medicine now' or you know, 'did you remember to take the pill' or something like that. . ." (Family 2, Female). Pepper might also help the young adult to do routine work every day. One caregiver said "Self-care, like brushing teeth, showering. . .she [young adult CR] can't differentiate 15 min. from an hour. Pepper can remind her after brushing for a few minutes" (Family 11 caregiver, female). Other caregivers also shared:

"Pepper could say, 'Hey M [young adult CR], today's the day we order our groceries' or 'Today's grocery day. Would you like to order groceries now?'. . .Pepper could. M does a really good job of brushing her teeth. But if something happens in our routine and it's distracting, M might forget how to brush her teeth" (Family 4 caregiver, male).

"[Pepper] could be reminders. . .D [young adult CR] doesn't like me to tell him things, because he wants to be more independent. But if Pepper was there telling [him] things, then he could listen to Pepper rather than me" (Family 9 caregiver, female).

**Theme four: Safety or monitoring.**    Overall, the majority of the caregivers indicated that Pepper could potentially provide safety or monitoring of their CRs if they had one at home in the future. One grandparent caregiver mentioned, "With the technology, you know, you could have respite and monitoring at the same time" (Family 5 caregiver, female). Other caregivers similarly expressed:

"It was exciting. . .Pepper could send me a signal that says M's having a migraine headache. . .Having that comfort, knowing that Pepper could ask a couple basic questions and send an alert, even a video, like a FaceTime connection" (Family 4 caregiver, male).

"Well, the fact that Pepper tracks her, I could feel comfortable leaving her at home by herself, knowing that in a way Pepper would be keeping an eye on her. She has seizures. So, I mean if she were to have a seizure, I would think that there'd be some way to program Pepper to detect that" (Family 11 caregiver, female).

**Theme five: Interactive engagement.**    The majority of the young adults with IDD experienced the powerful interactive engagement with Pepper and caregivers also reported observing positive interactions between the young adults and Pepper. Some parents expressed that they had never seen their child engage in daily life like they did during the program demonstration. One grandparent caregiver said, "It was really interesting. I liked the way he interacted" (Family 5 caregiver, female). Some participants were especially touched by the story that Pepper told the young adults with IDD as well as the interactions that Pepper asked them to share their own stories in school. One young adult said "It was amazing. And I thought the story. . .it was so, so, so emotional" (Family 1 young adult, female). And her mother also said that "It's hard to explain. . .just a sign of relief and knowing that even though Pepper's not real. . .this robot can interact with her" (Family 1 caregiver, female). Similarly, another family dyad was also moved by the story part, as the mother mentioned that "And if Pepper was there then he would be like he's got something that's occupying his mind and his time and is interacting with him you know" (Family 2 caregiver, female). Her young adult CR also said:

"It was amazing how he told his story, and you know, and how I told my story. . .I was actually excited because when I was looking at him, um, he kind of like read me, like an actual book" (Family 2 young adult, male).

Some young adults with IDD liked all the interactions, but also had favorites among specific programs. For example, one young adult told us:

"I liked interacting with Pepper. It was amazing. It was my first time interacting. . .It was awesome!. . .Air guitar [was awesome.]. . .and then Tai Chi. I felt like it was more relaxing with the Tai Chi and everything" (Family 11 young adult, female).

Some caregivers thought all the activities designed in the program helped to promote interaction and engagement between Pepper and their young adult CRs. One caregiver told us "I really liked fist bumping him. . .and probably a little bit of dancing. . .Well, I really loved how we did air guitar together" (Family 8 mom caregiver, female). Other family caregivers had similar feelings:

"It would give her something to interact with and to work her mind. . ..She loves books so having a story read to her. . .she loves to be read to and stuff. . .Even just like songs and stuff too. . .Maybe some songs she would like to sing, things like that she would also really enjoy interacting" (Family 7 caregiver, female).

"I mean, that she could interact, you know, with somebody that's not a parent. I mean, there's nurses and stuff, but the typical friends down the street and stuff like that, that is just not uh, there" (Family 1 caregiver, female).

One parent also thought such interactions and engagement may provide respite for them if they were away from their young adult CRs for a while:

"The physical activity, the music, the dance is going to be good. Because the last thing you want is if you're leaving someone you know for a few hours is for them to just sit around and not do anything" (Family 3 caregiver, female).

Another father caregiver had similar comment, but he also had some instructive suggestions for Pepper in the future:

"I saw an immediate level of excitement. I was sitting there in tears watching because I saw this sunshine, this flower, just kind of. . .Because she's been anticipating this. . .The interaction was very exciting and very uplifting, I think. Very positive. . .I saw the joy and the interaction with Pepper. . .If Pepper learned that about her personality, Pepper could make those suggestions. Like 'M [young adult CR], what would you like to watch on Netflix today? Is there a favorite movie you like? Engagements like this" (Family 4 caregiver, male).

## Discussion

The primary aim of this study was to explore the feasibility, acceptability, and usefulness of a novel program for providing respite care to older family caregivers of young adult CRs with IDD using a mixed methods experimental design. The study included three phases: focus groups with caregivers and their young adult CRs to inform the development of a program

with robot Pepper, surveys before and after the implementation of the program, and follow-up interviews with family dyads to capture their experiences with the program. Findings from the research in general suggested that the SAR (Pepper) engaged the young adults with IDD in a series of stimulating exercises and interactions, as well as provided potentials of physical/emotional respite for their older family caregivers.

The Phase I focus groups provided key input for the design of Phase II program implementation. The findings from Phase II showed both older caregivers and their young adult CRs scored the social presence of Pepper favorably and were satisfied with Pepper. Caregivers also reported positive perceptions of Pepper in terms of anthropomorphism, animacy, likeability, perceived intelligence, and perceived safety. These results were consistent with a previous study that showed older adults had positive reactions after exposure to SAR activities [52] as well as a systematic review that revealed older adults generally perceived robots positively and expected the robots to perform well at all times with few exceptions [53]. However, the previous literature focused on older adults in general, unlike the present study that focused on older caregivers of persons with IDD. The present study thus makes a unique contribution to the literature, owing to the fact that the older adults are in a caregiving role, rather than as companions or as young adults themselves.

In Phase II, no significant changes were found in caregivers' stress/burden, or well-being of young adults with IDD. The lack of significant findings might have been influenced by the "dosage" of the program (i.e., one time, only approximately 15 minutes) that may have been insufficient to impact the caregivers' burden or well-being. In fact, several studies of caregiver stress relief that provided more prolonged respite have shown minimal or no reduction of stress, attributed to the robustness of chronic stress patterns, while qualitative findings showing benefits were not captured in quantitative measures [54]. Hence, we included qualitative follow-up interviews in Phase III.

Content analysis from the phase III interviews suggested that the SAR may offer physical/emotional respite for caregivers because it has potentials of providing companionship/friendship as well as promoting independence, safety/monitoring, and interactive engagement with young adults. These findings were consistent with a previous qualitative study that showed children with disability and their parents perceived the SAR (in this case, NAO) to have therapeutic value through its potential to enhance engagement, promote child independence during rehabilitation exercises and its potential to support a rehabilitation program when a human therapist is not accessible [55]. Findings from Phase III underscore previous research suggesting that SAR could potentially enhance the well-being and decrease the workload on caregivers [33]. The positive reaction of caregivers in particular to Pepper was quite striking, which they described as being due to the way in which Pepper enabled new, or seldom seen behaviors on the part of their adult child. With behaviors and a storyline specifically designed to meet some of the caregiver and young adult's needs and aspirations, Pepper as a respite provider is quite out of the ordinary since Pepper's presence evokes strong positive feelings on the part of caregiver and young adult alike.

Because of challenging behaviors, speech impairment, physical disabilities, complex medical conditions, mental disorders and need for assistance services, young adults with IDD are at risk of social exclusion [14]. Social inclusion for people with IDD is an issue of emerging importance in research, policy, and practice [56]. For young adults with IDD, social inclusion is particularly important as they transition to environments outside the home, move into the community, and engage in formal programs aimed at developing independence skills as well as facilitating social interactions [57]. Interactions with Pepper were programed and scripted to provide social companionship/friendship, stimulating movements, and enjoyment formed the basis for engagement that made brief respite for the older adult caregiver possible.

In our study these interactions took place in the lab, and thus outside the home, rather than a community setting. A home setting would offer a more organic or natural environment for interacting with Pepper and provide more accessible respite for the older family caregiver. Community settings provide more opportunities for informal social interactions and inclusion., but the intentional engagement offered by Pepper, while not strictly speaking formal skills training for the young adults, nonetheless provided stimulation and learning generalizable to other settings outside the home. From an ethical standpoint, interactions with Pepper are not equivalent to social interactions with others in community settings, which have greater benefits to autonomy and self-determination, nor are they meant to be equivalent–just as the temporary respite offered by Pepper is not meant to stand in lieu of family caregiving. Rather, the emphasis is on providing a safe environment and privacy for engaging with Pepper, while respecting behavioral norms established by the family and young adult in the developmental stages of the robot respite scenario.

While a 'novelty effect' might account for some of their enthusiasm for Pepper in the absence of prior experience, the caregivers observing their young adult CRs' interactions with Pepper often remarked on specific behaviors, new to the young adult child, triggered by the demonstration in phase II. Pepper appeared to exceed the behavioral expectancies of the caregivers, and possibly the young adults with IDD, more than an equally unfamiliar human respite provider might have. This is interesting because in a previous study involving brief interactions with a human partner and Pepper, the researchers found a 'hyperpersonal' effect, in which people opened up to the robot more than to another human [58]. This ready self-disclosure is consistent with findings [59] that credibility might be given to an anthropomorphic computer without even conscious thought, based on interactions, for which a SAR like Pepper more than meets criteria. In other words, Pepper seemed to engage the young adults, and their observing caregivers, as a trustworthy and safe intelligent agent.

In reviewing our findings, it is important to consider this pilot study's limitations. First, the small sample size limited our analysis power to yield significant findings and representativeness. One eligibility criterion was the absence of cognitive impairment of young adults with IDD, which does not represent all young adults with IDD. A larger and more representative sample, for example, one that included persons with IDD from ethnically diverse backgrounds would shed light on implementation in the broader population. Furthermore, quality of life or caregiving stress/burden might not best capture the effectiveness of this 15-minute intervention. Future studies should consider selecting measures that are sensitive to change in the moment, such as calmness and mood. Second, regarding the older adult caregivers, a recent review of the effectiveness of SAR for older adults found a dearth of high-quality studies [60], suggesting the need for more robust sample sizes and methods in the extant research literature. Future studies might employ 'machine learning' [61]–in which Pepper would learn through social interaction without explicit instructions and use the experience to enhance performance without being programmed. Third, the interview results provide additional lessons learned for future SAR respite research. Caregivers spoke about safety and relationship norms lending credence to the notion that the normative preferences of older adults (i.e., caregivers) are closely linked to ethical concerns [53, 62]. Thus, in future studies, the normative expectations of caregivers around SAR respite should be direct inputs into the design process from the get-go, together with attention to an iterative ethics-based approach to design [63].

## Conclusion

By utilizing a mixed methods experimental design, this study conducted focus group interviews to inform the program development, pre and post survey to test the feasibility,

acceptability, and usefulness of the program implementation, as well as dyads interview to offer a more in-depth understanding of the findings. Results of the study demonstrated that programs with robotics could be a promising approach for filling a needed gap on designing caregiver programs for older family caregivers of persons with IDD that hold potential for translation, implementation, and sustainability.

Scientific and technical advances in robotics have been enormous and SARs are becoming more and more sophisticated. They can perform various daily tasks, have the ability to listen, talk and monitor the health status of persons with IDD, and to notify caregivers and professionals in case of an emergency. SAR thus has potential to be helpful companions for persons with IDD. Given the need for accessible IDD caregiver programs, this study offers an example of how to utilize SAR to provide respite care for caregivers of persons with IDD in the communities in which they live.

## Author Contributions

**Conceptualization:** Ling Xu, Noelle L. Fields, Julienne A. Greer, Priscila M. Tamplain, John C. Bricout, Bonita Sharma, Kristen L. Doelling.

**Formal analysis:** Ling Xu, Noelle L. Fields, Julienne A. Greer.

**Funding acquisition:** John C. Bricout.

**Methodology:** Ling Xu.

**Project administration:** Ling Xu, Noelle L. Fields, Julienne A. Greer, Priscila M. Tamplain.

**Software:** Kristen L. Doelling.

**Supervision:** Ling Xu, Julienne A. Greer.

**Writing – original draft:** Ling Xu, Noelle L. Fields, Julienne A. Greer, John C. Bricout, Bonita Sharma.

**Writing – review & editing:** Ling Xu, Noelle L. Fields, Julienne A. Greer, Priscila M. Tamplain, John C. Bricout.

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
