## [Decision Letter · Decision Letter 0]

10 May 2022

PONE-D-21-22611Supporting Older Family Caregivers of Young Adults with Intellectual and Developmental Disabilities (IDD): A Pilot Program with Socially Assistive RoboticsPLOS ONE

Dear Dr. Xu,

Thank you for submitting your manuscript to PLOS ONE. After careful consideration, we feel that it has merit but does not fully meet PLOS ONE’s publication criteria as it currently stands. Therefore, we invite you to submit a revised version of the manuscript that addresses the points raised during the review process.

We look forward to receiving your revised manuscript.

Kind regards,

Professor Benjamin Tan, BNSc MMed PhD RN

Journal Requirements:

3. Thank you for stating the following financial disclosure: "The contents of this article were developed in part under a grant from the National Institute on Disability, Independent Living, and Rehabilitation Research (NIDILRR grant number 90RE5025). NIDILRR is a Center within the Administration for Community Living (ACL), Department of Health and Human Services (HHS). The contents of this paper do not necessarily represent the policy of NIDILRR, ACL, HHS, and you should not assume endorsement by the Federal Government."

4. Thank you for stating the following in the Acknowledgments Section of your manuscript: "The contents of this article were developed in part under a grant from the National Institute on Disability, Independent Living, and Rehabilitation Research (NIDILRR grant number 90RE5025). NIDILRR is a Center within the Administration for Community Living (ACL), Department of Health and Human Services (HHS). The contents of this paper do not necessarily represent the policy of NIDILRR, ACL, HHS, and you should not assume endorsement by the Federal Government."

Please remove any funding-related text from the manuscript and let us know how you would like to update your Funding Statement. Currently, your Funding Statement reads as follows: "The contents of this article were developed in part under a grant from the National Institute on Disability, Independent Living, and Rehabilitation Research (NIDILRR grant number 90RE5025). NIDILRR is a Center within the Administration for Community Living (ACL), Department of Health and Human Services (HHS). The contents of this paper do not necessarily represent the policy of NIDILRR, ACL, HHS, and you should not assume endorsement by the Federal Government."

6. Please ensure that you refer to Figure 1 in your text as, if accepted, production will need this reference to link the reader to the figure.

Reviewers' comments:

Reviewer's Responses to Questions

**Comments to the Author**

1. Is the manuscript technically sound, and do the data support the conclusions?

Reviewer #1: Partly

2. Has the statistical analysis been performed appropriately and rigorously? 

Reviewer #1: Yes

3. Have the authors made all data underlying the findings in their manuscript fully available?

Reviewer #1: Yes

4. Is the manuscript presented in an intelligible fashion and written in standard English?

Reviewer #1: Yes

5. Review Comments to the Author

Reviewer #1: This is a very interesting paper and the authors are embarking on a very relevant topic. I appreciated the stages of the research study - first working with people with developmental disabilities and their parents to adapt the robot, and then doing a carefully designed experiment in the lab, and then evaluating it.

I think the study can be strengthened by shifting the language and describing what was done differently.

My most fundamental question is why is the study emphasizing supporting older family caregivers of young adults and not the value of robots for young people and their family caregivers? The authors do include the young adults in all stages of the study but yet they are secondary in the title and emphasis from research questions. A robot only ends up supporting social respite if first, fundamentally, the robot offers social companionship and entertainment to the individual. I think whether this might work to also provide respite becomes a secondary exploratory objective.

The experiment involves a 15 minute interaction with a novel machine. This does not "Test" respite at all. It is not in the home, it is not with something the young adult is familiar with and the parent is not going about other activities not watching the young person. This experiment is an early exploration or development of something that might be useful in this way. That needs to be more heavily emphasized.

Related to that, given that it is only 15 minutes in a lab environment outside of the home, with something novel the young person is not used to, the pre and post measures seem odd. They would not reflect changes in stress levels through having respite. They simply measure the reaction to the 15 minute experiment. It could be ok to still use these measures but they should be framed differently.

I do not think that this study looks at how much these robots help with social respite. They look at whether young adults and families like them and if they think they would be willing to explore how in the future, depending on the way they work, they could be an alternative to humans to provide social respite.

Also importantly, because all of the young people were able to provide their own consent, this suggests that the utility of these robots would be helpful with young people who are able to manage with some independence at home, emphasizing social aspects of respite, as opposed to safety issues. These were not young people with more severe disabilities requiring more constant hands on supervision.

A few additional specific comments follow:

I would remove their CR's from the manuscript and use terms that seem more empowering and respectful of these young people. Line 41 page 2 as an example. Could say instead: Both the young adults and their parents scored favorably... (note they are all parents according to the table). I would also remove the term "children" when possible to avoid confusion on the age of the young people.

I am not sure why in the first paragraph authors use the term intellectual and developmental disorder and right after developmental disabilities. Perhaps use disabilities instead of disorder throughout. (p 3 line 60).

I would revise the introduction to talk about the importance of companionship and support for young people with IDD living with older parents and the possibility of SAR. The examples of SAR provided with older people are specific to the older individual who needs social assistance and not to the caregivers of these older people and the same should be for those with IDD. There can be some discussion of need for respite but I would first talk about need for engagement and companionship to the individual.

In the literature on interventions for family caregivers, authors could cite literature reviews on mindfulness based interventions for family caregivers, and also on family caregiving of adults with IDD.

Line 97 page 4 - I am not sure the main reason for in home respite is financial. Paying someone to work at home and out of the home can cost the same amount. It may be easier for the family if it is at home and does not require the parent to take the person to respite, and it also may be more comfortable for the individual.

lines 109-115 page 5 on SAR focus on benefits to older person. So not a strong argument why SAR good for social respite to parent. This framing feels disrespectful to me.

I may change the objectives in this way:

1 development of SRA that was well accepted by both young adults with IDD and their aging parents..

2. development of adequate responsiveness and engagement with young adults

3. potential for improving well being of young adults

4. potential to provide respite and improve well being of older family caregivers

methods

page 7 line 150 is an example of their CR's, same line - make sure they understood, and line 151 in written FORMAT respectively

page 7 line 158 - comment on skills/abilities of individuals who participated does not represent all adults with IDD as they were all able to provide written informed consent independently.

Measures - not surprising that there was no change in quality of life and burden related to a brief 15 minute intervention. I would not have chosen these measures. For QofL for adults with IDD, is this measure valid for these young adults? My guess is it would not be appropriate.

This will ultimately be the authors' decision but I do not think that the parent or young adult pre post measures add anything to the study. I would have selected measures that are sensitive to change in the moment, such as calmness, mood. And for the young people I would use valid measures for them. I the qualitative comments are much more relevant, as well as measures of the experience with the robot.

Results

I found the quotes quite rich and informative. I might separate how people experienced the robot from what they think they could do with a robot if they had one in their home (either to interact, or for skills teaching or respite).

Discussion

I would reframe the discussion to de-emphasize respite (it did not test respite), but rather an exploration of whether SAR is feasible for companionship, skill building and potential respite from the perspectives of young adults and older family caregivers.

There should be some discussion on why young people need additional social companionship and also the ethical implications of supporting older families but providing robots in the home as opposed to more community based engagement opportunities.

Limitations

Not only was the representation of the older caregivers limited so was the representation of the young adults. This should be mentioned, especially around their cognitive ability. If future work wants to focus on respite, there needs to be more specific measurement of respite needs and what respite is.

6. PLOS authors have the option to publish the peer review history of their article (what does this mean?). If published, this will include your full peer review and any attached files.

Reviewer #1: No

---

## [Author Response · Author response to Decision Letter 0]

17 Jun 2022

We appreciate the reviewer’s valuable comments. We have highlighted our major changes in yellow color in the revised manuscript when possible.

Review Comments to the Author

This is a very interesting paper and the authors are embarking on a very relevant topic. I appreciated the stages of the research study - first working with people with developmental disabilities and their parents to adapt the robot, and then doing a carefully designed experiment in the lab, and then evaluating it.

1. I think the study can be strengthened by shifting the language and describing what was done differently. My most fundamental question is why is the study emphasizing supporting older family caregivers of young adults and not the value of robots for young people and their family caregivers? The authors do include the young adults in all stages of the study but yet they are secondary in the title and emphasis from research questions. A robot only ends up supporting social respite if first, fundamentally, the robot offers social companionship and entertainment to the individual. I think whether this might work to also provide respite becomes a secondary exploratory objective.

Response: Thank you for this comment. As mentioned in the literature, there has been a few studies used SAR to help young people or children with IDD. However, the use of SAR as a tool for respite, especially for older family caregivers of young adults with IDD remains under-studied. The present study aimed to address such literature gap. However, we appreciate your insight and revised the objectives of our study (page 7) to emphasize the importance of the young adults’ perspectives (e.g., responsiveness, engagement, potential for improving their well-being). We also revised our title to illustrate the potential value of the SAR for both the young adults and the older family caregivers. 

2. The experiment involves a 15 minute interaction with a novel machine. This does not "Test" respite at all. It is not in the home, it is not with something the young adult is familiar with and the parent is not going about other activities not watching the young person. This experiment is an early exploration or development of something that might be useful in this way. That needs to be more heavily emphasized.

Response: Yes, I agreed with the reviewer that the present study is in early exploratory and developmental stage. This pilot study offers promising for respite care to family caregivers. A future study with large sample size and robust study design (e.g., randomization, control and experimental groups) is needed to test whether this program effectively work. We mentioned such limitations and future study on page 26. 

3. Related to that, given that it is only 15 minutes in a lab environment outside of the home, with something novel the young person is not used to, the pre and post measures seem odd. They would not reflect changes in stress levels through having respite. They simply measure the reaction to the 15-minute experiment. It could be ok to still use these measures but they should be framed differently.

Response: Thank you for this comment. We agree with the reviewer that this 15-minute trial program is too short to test its effect on the stress level, which we acknowledged this limitation on page 26. Partially because of the short dosage of intervention, some outcome measures from the pre and post tests among older caregivers (self-rated health and caregiving stress/burden) and children with IDD (quality of life) were not statistically significant. But the study did measures reflections to the 15-minute experience, such as social presence of robot, social engagement, perceptions of robot and satisfaction of interaction with robot (see Table 3). 

As the first step of a novel intervention study, we had to implement this program in a lab. However, this ‘research room’ space is dedicated to providing a living area for research studies to simulate the home-space of participants. In this way, participants receive a home-like environment while interacting with the social robot and reimagine how that robot might interact with their real home life (see more details on pages 10-11). We also acknowledged this limitation on page 26. 

4. I do not think that this study looks at how much these robots help with social respite. They look at whether young adults and families like them and if they think they would be willing to explore how in the future, depending on the way they work, they could be an alternative to humans to provide social respite.

Response: The primary goal of this study is to test if robot Pepper can provide with respite for older family members, and the secondary goal is to test the acceptance and useless of robot in providing respite. We clarified this on page 7 in the revised manuscript. Therefore, this study used a mixed methods experimental design (Creswell & Clark, 2017) in which we collected the qualitative data before the intervention (e.g., Phase 1, focus groups) to help with the development of the project (Phase 2, intervention) as well as after the intervention to enrich the interpretation of the experimental results (e.g., Phase 3, individual interviews). 

Creswell, J. W., & Clark, V. L. P. (2017). Designing and conducting mixed methods research. Sage publications.

5. Also importantly, because all of the young people were able to provide their own consent, this suggests that the utility of these robots would be helpful with young people who are able to manage with some independence at home, emphasizing social aspects of respite, as opposed to safety issues. These were not young people with more severe disabilities requiring more constant hands-on supervision.

Response: Though one eligibility criteria is their absence of cognitive impairment, some (approximately half) of the young adults need to understand the consent form under help of their older family caregivers, especially for those with Down Syndrome. We revised this on page 8. We also acknowledged this limitation of the sample representativeness on page 26.

6. A few additional specific comments follow:

I would remove their CR's from the manuscript and use terms that seem more empowering and respectful of these young people. Line 41 page 2 as an example. Could say instead: Both the young adults and their parents scored favorably... (note they are all parents according to the table). I would also remove the term "children" when possible to avoid confusion on the age of the young people.

Response: The majority of the caregivers are parents, except one family. Caregiver of Family 5 is grandparent. We replaced “caregivers and their CRs” with “young adults and their caregivers” throughput the text in the revised manuscript. In order not to lose the caregiving relationship, we added these sentences in the introduction part “The caregivers and care recipients in this study are co-community dwelling, in which the caregiver is the parent or grandparent, and the care recipient is the young adult with IDD. Thus, we will refer to the care recipient as a young adult with IDD, acknowledging that they live together in a home/familial setting.”

7. I am not sure why in the first paragraph authors use the term intellectual and developmental disorder and right after developmental disabilities. Perhaps use disabilities instead of disorder throughout. (p 3 line 60).

Response: We replaces “disorder” with “disabilities” as suggested.

8. I would revise the introduction to talk about the importance of companionship and support for young people with IDD living with older parents and the possibility of SAR. The examples of SAR provided with older people are specific to the older individual who needs social assistance and not to the caregivers of these older people and the same should be for those with IDD. There can be some discussion of need for respite, but I would first talk about need for engagement and companionship to the individual.

Response: Thanks for this suggestion. We added the importance of engagement, companionship and support for young adults with IDD living with older parents and the possibility of SAR. See yellow highlights on page 4. 

9. In the literature on interventions for family caregivers, authors could cite literature reviews on mindfulness based interventions for family caregivers, and also on family caregiving of adults with IDD.

Response: Thanks for this suggestion. Because of the page limit of the journal as well as our literature focus on the respite care program/intervention, we only used two sentences to summarize mindfulness-based intervention on page 5 “Most programs designed to address the mental health of caregivers come in the form of stress reducing intervention. For example, a community-based intervention called Mindfulness Based Stress Reduction (MBSR) used mindfulness and demonstrated decreased stress and anxiety among caregivers”. We added “Respite” to the subheading of “programs or intervention for family caregivers of persons with IDD”, so that the focus of the literature is much clearer. 

10. Line 97 page 4 - I am not sure the main reason for in home respite is financial. Paying someone to work at home and out of the home can cost the same amount. It may be easier for the family if it is at home and does not require the parent to take the person to respite, and it also may be more comfortable for the individual.

Response: We stated that “there may be some financial supports available for these services. These financial supports can come from Medicaid or other forms of ongoing health insurance provide to children and youth with special needs”. Therefore, financial support is not the main reason. We revised this sentence on page 5 (line 110-113) to make it clearer. 

11. lines 109-115 page 5 on SAR focus on benefits to older person. So not a strong argument why SAR good for social respite to parent. This framing feels disrespectful to me.

Response: Thank you for this comment. We want to ensure that we communicate respect to all of our readers. In this paragraph, we summarized the overall benefits of SAR, its benefit to children with disabilities, and benefits to the caregiving process (e.g., dementia caregiving). Based on the literature on SAR’s benefits, we hypothesized similar benefits for older family caregivers of young adults with IDD. However, the use of SAR as a tool for respite, especially for older family caregivers of young adults with IDD remains under-studied (last sentence of this paragraph). To make this argument clearer, we added a few more sentences on page 6 in the revised manuscript (lines 131-140). 

12. I may change the objectives in this way:

1 development of SRA that was well accepted by both young adults with IDD and their aging parents.

2. development of adequate responsiveness and engagement with young adults

3. potential for improving well being of young adults

4. potential to provide respite and improve well being of older family caregivers

Response: Thank you for this suggestion. We revised accordingly. See yellow highlights on page 7 (lines 154-158).

13. methods

page 7 line 150 is an example of their CR's, same line - make sure they understood, and line 151 in written FORMAT respectively

page 7 line 158 - comment on skills/abilities of individuals who participated does not represent all adults with IDD as they were all able to provide written informed consent independently.

Response: We replaced “their CRs” throughout the text when needed. We also added “format” after “in written”. Yes, this criterion makes our sample does not represent all adults with IDD. We added this limitation on page 26. Please also see our response to comment #5 above.

14. Measures - not surprising that there was no change in quality of life and burden related to a brief 15-minute intervention. I would not have chosen these measures. For QofL for adults with IDD, is this measure valid for these young adults? My guess is it would not be appropriate.

This will ultimately be the authors' decision but I do not think that the parent or young adult pre post measures add anything to the study. I would have selected measures that are sensitive to change in the moment, such as calmness, mood. And for the young people I would use valid measures for them. I the qualitative comments are much more relevant, as well as measures of the experience with the robot.

Response: The Quality of Life (QoL) of young adults with IDD was measured through QoL Integral Scale. The psychometric properties of the QOL Integral Scale indicate that the instrument is reliable and valid among persons with IDD (Verdugo, Gómez, Arias, & Schalock, 2010), which we described on pages 13-14. We agreed with the reviewer that calmness or mood might be more sensitive to catch the change in the moment than the quality of life or burden. We added this in the limitation part on page 26 and will add these measures to the future study. Thank you for this suggestion. 

Verdugo MÁ, Gómez LE, Arias B, Schalock RL. The Integral quality of life scale: development, validation, and use. In Enhancing the quality of life of people with intellectual disabilities 2010 (pp. 47-60). Springer, Dordrecht.

15. Results

I found the quotes quite rich and informative. I might separate how people experienced the robot from what they think they could do with a robot if they had one in their home (either to interact, or for skills teaching or respite).

Response: Thank you for this suggestion. After re-read this qualitative result part, we found that it is hard to separate how caregivers experienced the robot from what they think could do with a robot if they had one in their home. Among the five themes identified, “interactive engagement with their children” and “providing companionship/friendship” are clearly from caregivers’ observation or young adults’ own experience. The other three themes, physical and/or emotional respite, promoting independence and safety or monitoring, are by large part what the participants think SAR could do potentially. In the revised manuscript, we added 1-2 sentences under each theme to make it much clearer whether the theme was based on their actual observation/experience or their thoughts that SAR could offer in the future.

16. Discussion

I would reframe the discussion to de-emphasize respite (it did not test respite), but rather an exploration of whether SAR is feasible for companionship, skill building and potential respite from the perspectives of young adults and older family caregivers. There should also be some discussion on why young people need additional social companionship and also the ethical implications of supporting older families but providing robots in the home as opposed to more community-based engagement opportunities.

Response: We appreciate the reviewer’s comment to de-emphasize respite. The whole project was funded and developed for providing respite care. Therefore, the primary aim of this study was to explore its feasibility, acceptability, and usefulness in providing temporary respite care. Engagement of the young adult in an ethical, safe and stimulating interactive scenario with Pepper, scripted and programed with feedback from the young adults and their family members was foundational to the temporary respite. Temporary respite was operationalized in the context of the social learning and interaction scenario with Pepper. Even though no statistical significance was found for the well-being outcomes, partially due to the short ‘dosage’ of the program (e.g., 15 minutes), the study showed that SAR is perceived as providing companionship, skill building, interactive engagement, while promoting independence, and safety or monitoring. Through these mechanisms, this program has potential in providing physical and/or emotional respite for older family caregivers in tandem with a stimulating and engaging social interactions for the young adults. We have revised the discussion carefully to make ensure that the study rationale and aims are clearer.

As suggested, we provided additional narrative in the introduction, as well as in the discussion, that notes the importance of extra-familial social interactions for young adults with IDD in service of more companionship and social inclusion, as well as greater independence (autonomy). We also reflected briefly on the ethical implications of different settings (home, lab, and community) for situating social interactions with robots, and their implications for the independence, autonomy, and social connectedness of young adults with intellectual disabilities.

17. Limitations

Not only was the representation of the older caregivers limited so was the representation of the young adults. This should be mentioned, especially around their cognitive ability. If future work wants to focus on respite, there needs to be more specific measurement of respite needs and what respite is.

Response: Thank you for this comment. I added the limitation of the criterion of absence of cognitive impairment on page 26. Please also see our response to above comments #5 and #13.

---

## [Editor Report · Decision Letter 1]

10 Aug 2022

Socially Assistive Robotics and Older Family Caregivers of Young Adults with Intellectual and Developmental Disabilities (IDD): A Pilot Study Exploring Respite, Acceptance, and Usefulness

PONE-D-21-22611R1

Dear Dr. Xu,

We’re pleased to inform you that your manuscript has been judged scientifically suitable for publication and will be formally accepted for publication once it meets all outstanding technical requirements.

Kind regards,

Robert Didden

Academic Editor

PLOS ONE
---

## [Editor Report · Acceptance letter]

2 Sep 2022

PONE-D-21-22611R1 

Socially Assistive Robotics and Older Family Caregivers of Young Adults with Intellectual and Developmental Disabilities (IDD): A Pilot Study Exploring Respite, Acceptance, and Usefulness 

Dear Dr. Xu:

I'm pleased to inform you that your manuscript has been deemed suitable for publication in PLOS ONE. Congratulations! Your manuscript is now with our production department. 

Kind regards, 

on behalf of

Professor Robert Didden 

Academic Editor

PLOS ONE